# Comparative Morphological, Ultrastructural, and Molecular Studies of Four Cicadinae Species Using Exuvial Legs

**DOI:** 10.3390/insects10070199

**Published:** 2019-07-06

**Authors:** Jun-Ho Song, Wook Jin Kim, Ji-Min Cha, Sungyu Yang, Goya Choi, Byeong Cheol Moon

**Affiliations:** Herbal Medicine Resources Research Center, Korea Institute of Oriental Medicine, Naju 58245, Korea

**Keywords:** barcoding, Cicadidae Periostracum, *Cryptotympana atrata*, insect leg, SEM, exuviae

## Abstract

Previous studies have suggested that exuviae can be used for the identification of cicada species, but the precise characteristics that differ among species have not been determined. Thus, we performed the first comparative analyses of the leg morphology, ultrastructure, and mitochondrial DNA sequences of exuviae of four dominant cicada species in Korea, *Hyalessa maculaticollis* (Motschulsky, 1866), *Meimuna opalifera* (Walker, 1850), *Platypleura kaempferi* (Fabricius, 1794) and *Cryptotympana atrata* (Fabricius, 1775), the source of Cicadidae Periostracum, a well-known traditional medicine. A morphological analysis revealed that the profemur length, femoral tooth angle, and distance between the intermediate and last tooth of the femoral comb are useful characteristics for identification. We also evaluated the usefulness of the size, degree of reflex, and number of spines on the mid-legs and hind legs as diagnostic features. An ultrastructural study showed that *Meimuna opalifera* has a unique surface pattern on the legs. The sequences obtained using exuviae were identical to previously obtained sequences for adult tissues. Moreover, in a phylogenetic analysis using *CO1* sequences, each species formed a monophyletic cluster with high bootstrap support. Accordingly, multiple methodological approaches using exuviae might provide highly reliable identification tools. The integrative data provide useful characteristics for the exuviae-based identification of closely related species and for further taxonomic and systematic studies of Cicadinae.

## 1. Introduction

Cicadas (Hemiptera, Cicadidae) are plant-sucking insects characterized by loud, species-specific calling songs produced by adult males during the summer [1]. Their immature life stage, which is much longer than the adult stage, continues for one to 17 years underground [2,3,4,5,6]. Cicadas must periodically shed their exoskeleton for growth [5,7]. Exuviae are the remains of exoskeletons after ecdysis (molting) of final instar nymphs and are associated with adult structures [8].

Exuviae of cicadas are commonly used in traditional medicine in Asia, including Korea, China, and Taiwan. In the Korean Herbal Pharmacopoeia, authentic Cicadidae Periostracum refers to the exuviae of *Cryptotympana dubia* Haupt, 1917 or *Cryptotympana pustulata* Stål, 1861, belonging to the family Cicadidae [9,10], both of which are synonyms of *Cryptotympana atrata* (Fabricius, 1775). Thus, the only source of Cicadidae Periostracum is *C. atrata* [11,12]. Pharmacopoeia of the People’s Republic of China and the Taiwan Herbal Pharmacopoeia have also designated the exuviae of *C. atrata* [=*C. pustulata*] as authentic Cicadidae Periostracum [11,13,14].

In traditional Chinese medicine, Cicadidae Periostracum has been used for clinical purposes owing to its anticonvulsive, sedative, and hypothermic effects [15]. Modern pharmacological studies of Cicadidae Periostracum have shown that it has fibrinolytic [16] and anti-bacterial activity [17] as well as anti-inflammatory [18] and anti-allergic effects [19]. To verify its therapeutic effects and safety, accurate authentication is critical. Thus, various and multilateral methods for the identification and authentication of the source species are essential.

The morphological identification of final instar nymphs or exuviae of cicadas has been evaluated, in addition to pharmacological aspects [12,20,21,22,23,24,25,26]. Recently, studies of cicadas have focused on the morphometrics of exuviae in central Korea [27] and of final instar nymphs in China [28] for accurate identification. Based on descriptions of final instar nymphs or exuviae in cicadas, characteristics of the fore-, mid-, or hind legs may have taxonomic or phylogenetic importance [2,3,25,26,29,30]. However, comprehensive morphological and ultrastructural analyses focused on all leg parts, including the femur, femoral comb, tibia, and tarsus, are lacking.

Recently, exuviae have been used for molecular analyses of various species [31,32,33,34]. Genomic DNA from muscle tissues or metabolic products on the inner side of the exoskeleton can be extracted and analyzed [35]. However, very few molecular studies have evaluated exuviae of cicadas [8,36]. Furthermore, multiple methodological approaches, combining morphological and molecular data, are rarely used to evaluate cicadas.

*Hyalessa maculaticollis* (Motschulsky, 1866) (tribe Sonatini Lee, 2010), *Meimuna opalifera* (Walker, 1850) (tribe Dundubiini Distant, 1905), *Platypleura kaempferi* (Fabricius, 1794) (tribe Platypleurini Schmidt, 1918), and *C. atrata* (tribe Cryptotympanini Handlirsch, 1925), all belonging to the subfamily Cicadinae Latreille, 1802, are the dominant cicadas in Korea. Moreover, Cicadidae Periostracum, which is exuviae of *C. atrata*, as an insect medicinal source can potentially be adulterated and misused with the exuviae of these dominantly distributed species.

Thus, the main aims of the present study were to (1) provide detailed descriptions and illustrations of the exuvial leg morphology and ultrastructures of four cicada species, using a comparative approach, and develop a taxonomic key; (2) compare mitochondrial DNA sequences and evaluate phylogenetic relationships; and (3) evaluate the correlation between morphological and molecular data. This study is the first step towards determining the conserved and useful characteristics of exuvial legs. Our findings show that microscopy and DNA barcoding are valuable tools for the accurate identification and effective quality control of insect medicinal materials.

## 2. Materials and Methods

### 2.1. Materials

Exuviae were collected from branch or trunk of trees (i.e., *Chionanthus retusus* Lindl. and Paxton (Retusa fringetree), *Prunus serrulata* f. *spontanea* (E.H. Wilson) C.S. Chang (Oriental flowering cherry), *Zelkova serrata* (Thunb.) Makino (Sawleaf zelkova), and other kinds of trees in Korean natural populations after molting from July to August 2018. The morphological characteristics of exuvial fore-legs, mid-legs and hind legs of these species were determined.

All materials were deposited in the Korean Herbarium of Standard Herbal Resources (Index Herbariorum code KIOM) at the Korea Institute of Oriental Medicine, Naju, Korea; *Cryptotympana atrata* (2-19-0006), *Hyalessa maculaticollis* (2-19-0007), *Meimuna opalifera* (2-19-0131), and *Platypleura kaempferi* (2-19-0130). Marshall et al. [37] was used as a reference to determine the correct names of all species.

### 2.2. Morphological Analysis

Morphological characteristics were observed using a stereomicroscope (Olympus SZX16; Tokyo, Japan). Morphological images were captured using a digital camera attached to the microscope (Olympus DP21, Olympus, Tokyo, Japan).

Twenty individuals of each of the four species were used to obtain measurements, for a total of 80 measurements for the fore-, mid-, and hind legs. The following parameters were evaluated: profemur length (FL), protibia length (FTL), femoral tooth angle (FA), number of femoral comb teeth (NFC), distance between the intermediate tooth of the femur and last tooth of the femoral comb (DIF), pretarsus length (FSL), mesotibia length (MTL), mesotarsus length (MSL), number of mesotibial spines (NMS), metatibia length (HTL), metatarsus length (HSL), and number of metatibial spines (NHS).

Most of these measurements were obtained using simple measurement tools (Olympus DP21 digital photography system, Olympus, Tokyo, Japan) based on images obtained using a digital camera (Olympus DP21) for microscopes. A set of digital Vernier calipers (CD-15CP; Mitutoyo, Kawasaki, Japan) was also used for measurements.

The terminology for structural features followed Midgley et al. [26] and Hou et al. [27].

### 2.3. PCA

A principal component analysis (PCA) was conducted to determine whether the quantitative morphological data allowed the grouping of species. The first three principal components with eigenvalues above 1.0 were obtained. These analyses were implemented in PC-ORD version 5.31 [38]. The PCA included twelve variables (FL, FTL, FA, NFC, DIF, FSL, MTL, MSL, NMS, HTL, HSL and NHS).

### 2.4. Ultrastructural Analysis

The fore-legs, mid-legs, and hind legs were excised from the exuviae and washed in 70% ethanol, and all samples were allowed to air-dry. Tarsi and tibia from all three legs and femurs from fore-legs were mounted directly on aluminum stubs using double-sided adhesive carbon tape. Stubs were coated with gold using a sputter coater (208HR; Cressington Scientific Instruments Ltd., Watford, UK), and all samples were observed using a low-voltage field emission scanning electron microscope (JSM-7600F; JEOL, Tokyo, Japan) at an accelerating voltage of 5 kV with a working distance of 8–10 mm. To confirm the consistency of morphological characteristics, at least three samples from each species were evaluated.

### 2.5. Preparation of Genomic DNA

Samples were identified based on morphological characteristics and genomic DNA was extracted from most samples (Appendix A). Genomic DNA was isolated according to the manufacturer’s protocol using the DNeasy^®^ Blood and Tissue Kit (Qiagen, Valencia, CA, USA). The purity and concentration of DNA were assessed using a spectrophotometer (Nanodrop ND-1000; Wilmington, DE, USA) and 1.5% agarose gel electrophoresis [39]. The final DNA concentration used for PCR amplification was approximately 15 ng/μL in TE buffer. Extracted DNA samples were stored at −20 °C.

### 2.6. PCR Amplification of CO1

Using the primers (*CO1*-C02 5′-AYT CAA CAA ATC ATA AAG ATA TTG G-3′ and *CO1*-C04 5′-ACY TCR GGR TGA CCA AAA AAT CA-3′) developed by Che et al. [40], a fragment of mitochondrial cytochrome oxidase subunit 1 was amplified. PCR amplifications were performed in 40 μL reaction volumes containing 10 mmol L^−1^ Tris-HCl (pH 9.0), 2.5 mmol L^−1^ MgCl_2_, 200 μmol L^−1^ each dNTP, 10 mmol L^−1^ (NH_4_)_2_SO_4_, 0.5 U *Taq* DNA polymerase (Solgent, Daejeon, Korea), 0.5 μmol L^−1^ each primer, and approximately 15 ng template DNA [39].

PCR amplification was performed using a DNA Engine Dyad^®^ PTC-0220 (Bio-Rad, Foster City, CA, USA). The following modified parameters of Kim et al. [39] were used: 95 °C for 5 min; 35 cycles of 1 min at 95 °C, 1 min at 45 °C and 1 min at 72 °C; and a final extension for 5 min at 72 °C. PCR products were separated by 1.5% agarose gel electrophoresis with a 100-bp DNA ladder (Solgent).

### 2.7. Nucleotide Sequence and Phylogenetic Analysis

Amplified mitochondrial *CO1* DNA fragments were extracted from agarose gels using a Gel Extraction Kit (Qiagen) and sub-cloned into the pGEM-T Easy Vector (Promega, Madison, WI, USA). Inserted fragments were sequenced in both directions using an automatic DNA sequence analyzer (ABI 3730; Applied Biosystems Inc., Foster City, CA, USA) as described by Kim et al. [39]. The samples of *C. atrata* (14 individuals: abbreviation CA), *H. maculaticollis* (17 individuals: HM), *M. opalifera* (eight individuals: MO), and *P. kaempferi* (nine individuals: PK) were used in the analysis (Appendix). The following *CO1* sequences for the four species deposited in NCBI GenBank were also analyzed: *C. atrata*, MG737717; *H. maculaticollis*, KY860344; *M. opalifera*, GQ527088; and *P. kaempferi*, MG737816.

Approximately 700-bp *CO1* sequences were edited using BioEdit version 7.2.5 [41]. The contigs were aligned to analyze intra- and interspecific sequence variation. For the analysis of sequence identity and divergence, inter- or intraspecific genetic distances were calculated using the Kimura-2-parameter (K2P) model in MEGA 6. A phylogenetic analysis based on the full-length *CO1* sequences was performed using MEGA 6 version 6.06 [42,43]. The phylogenetic tree was constructed using the NJ method with the K2P model, pairwise deletion for gaps/missing data treatment and 1000 bootstrap replicates, setting *Anthocharis cardamines* (Linnaeus, 1758) (MH420365) as an outgroup.

## 3. Results

We characterized morphological variation in the three leg types (fore-legs, mid-legs, and hind legs) in the four species. The major leg characteristics are summarized in Table 1. Representative legs of the four species are illustrated in Figure 1, Figure 2, Figure 3, Figure 4 and Figure 5.

### 3.1. Morphology and Ultrastructure of Three Legs from Four Cicada Exuviae

*Cryptotympana atrata* (Fabricius, 1775). Generally brown. Femur with posterior tooth long and sharp, curved forward slightly, length approximately two-times longer than the width of its base; accessory tooth small and blunt; intermediate tooth present (Figure 1a); distance between itf and ltf about two times longer than DIF (Table 1); surface of profemur with echinate ornamentation (Figure 3b); femoral comb usually with seven teeth (4/20 individuals with six and 1/20 with eight teeth), the first tooth about as large as or two-times wider than the second tooth (femoral formula 2–1–7) (Table 1), surface smooth (Figure 3a). Protibia arched, laterally flattened; apical tooth long, somewhat blunt; point of the blade of the tibia large and long, tooth-like, separated from the apical tooth of the blade by a strong incision (Figure 3c). Pretarsus well-developed, folded over the inner surface of the protibia, two-segmented; apical tarsomere elongated; pretarsal claws of unequal size (Figure 3d). Mesotibia with five, short apical spines (Figure 2a; Figure 4a), surface smooth with sparse setae (Figure 4b). Mesotarsus smooth with sparse setae (Figure 4c), apex two-segmented; apical tarsomere elongated; mesotarsal claws of unequal size (Figure 4d). Metatibia with six or seven apical spines (Figure 2e; Figure 5a), one much longer than the others, surface smooth with sparse setae (Figure 5b). Metatarsus smooth with sparse setae (Figure 5c), apex two-segmented; apical tarsomere elongated; metatarsal claws of unequal size (Figure 5d).

Measurements (Table 1): profemur length, 5.3–6.8 mm; protibia length, 5.6–6.9 mm; femoral tooth angle, 66.5–83.9°; number of femoral comb teeth, 6–8; distance between itf and ltf, 1.4–1.8 mm; pretarsus length, 3.9–4.7 mm; mesotibia length, 7.0–8.0 mm; mesotarsus length, 3.4–4.9 mm; number of mesotibial spines, 5; metatibia length, 7.1–8.1 mm; metatarsus length, 2.9–3.9 mm; number of metatibial spines, 6–7.

*Hyalessa maculaticollis* (Motschulsky, 1866). Generally dark brown. Femur with posterior tooth long and sharp, curved forward slightly, approximately two-times longer than the width of its base; accessory tooth small and blunt; intermediate tooth present (Figure 1b); distance between itf and ltf about four times longer than DIF (Table 1); surface of the profemur smooth with ornamentation (Figure 3f); femoral comb usually with seven teeth (2/20 individuals with six teeth), first tooth about two times wider than the second tooth (femoral formula 2–1–7) (Table 1), surface smooth (Figure 3e). Protibia arched, laterally flattened; apical tooth long, somewhat blunt; point of the blade of the tibia very small or almost absent (Figure 3g). Pretarsus well-developed, folded over the inner surface of the protibia, two-segmented; apical tarsomere elongated; pretarsal claws of unequal size (Figure 3h). Mesotibia with five apical spines, one much longer than the others (Figure 2b; Figure 4e), surface smooth with sparse setae (Figure 4f). Mesotarsus smooth with sparse setae (Figure 4g), apex two-segmented; apical tarsomere elongated; mesotarsal claws of unequal size (Figure 4h). Metatibia with five apical spines, one much longer than the others, inner surface of apical spines with setae (Figure 2f; Figure 5e), surface of metatibia smooth with sparse setae (Figure 5f). Metatarsus smooth with sparse setae (Figure 5g), apex two-segmented; apical tarsomere elongated; metatarsal claws of unequal size (Figure 5h).

Measurements (Table 1): profemur length, 5.4–6.7 mm; protibia length, 5.4–6.5 mm; femoral tooth angle, 59.9–77.4°; number of femoral comb teeth, 6–7; distance between itf and ltf, 0.5–0.8 mm; pretarsus length, 3.1–4.0 mm; mesotibia length, 7.2–8.9 mm; mesotarsus length, 2.7–3.8 mm; number of mesotibial spines, 5; metatibia length, 6.9–9.0 mm; metatarsus length, 2.3–2.9 mm; number of metatibial spines, 5.

*Meimuna opalifera* (Walker, 1850). Generally light yellowish brown. Femur with posterior tooth long and sharp, curved forward slightly, approximately two-times longer than the width of its base; accessory tooth small and blunt; intermediate tooth present (Figure 1c); distance between itf and ltf about four times longer than DIF (Table 1); surface of profemur echinate ornamentation (Figure 3j); femoral comb usually with eight teeth (2/20 individuals with seven teeth), the first tooth about two-times wider than the second tooth (femoral formula 2–1–8) (Table 1), surface smooth (Figure 3i). Protibia arched, laterally flattened; apical tooth long, very sharp; point of the blade of tibia short, tooth-like, separated from the apical tooth of the blade by a weak incision (Figure 3k). Pretarsus well-developed, folded over the inner surface of the protibia, two-segmented; apical tarsomere elongated; pretarsal claws of unequal size (Figure 3l). Mesotibia with five apical spines, nearly equal in length, inner surface of apical spines with setae (Figure 2c; Figure 4i), surface of mesotibia echinate with dense setae (Figure 4j). Mesotarsus echinate with sparse setae (Figure 4k), apex two-segmented; apical tarsomere elongated; mesotarsal claws of unequal size, echinate between two claws (Figure 4l). Metatibia with five apical spines, nearly equal in length, inner surface of apical spines with setae (Figure 2g; Figure 5i), surface of metatibia echinate with dense setae (Figure 5j). Metatarsus echinate with sparse setae (Figure 5k), apex two-segmented; apical tarsomere elongated; metatarsal claws of unequal size, echinate between two claws (Figure 5l).

Measurements (Table 1): profemur length, 3.6–4.8 mm; protibia length, 3.7–4.3 mm; femoral tooth angle, 50.3–64.0°; number of femoral comb teeth, 7–8; distance between itf and ltf, 0.4–0.7 mm; pretarsus length, 1.9–3.0 mm; mesotibia length, 5.2–7.3 mm; mesotarsus length, 1.7–2.6 mm; number of mesotibial spines, 5; metatibia length, 5.7–7.3 mm; metatarsus length, 1.7–2.2 mm; number of metatibial spines, 5.

*Platypleura kaempferi* (Fabricius, 1794). Generally yellowish brown. Femur with posterior tooth long and sharp, curved forward very slightly, length approximately three-times longer than the width of its base; accessory tooth small and blunt; intermediate tooth present (Figure 1d); distance between itf and ltf about as long as the DIF (Table 1); surface of profemur smooth (Figure 3n); femoral comb usually with seven teeth (all of the individuals with seven teeth), the first tooth about two- or three-times wider than the second tooth (femoral formula 2–1–7) (Table 1), surface smooth (Figure 3m). Protibia slightly arched, laterally flattened; apical tooth very short, blunt; point of the blade of tibia very short, small, tooth-like, separated from the apical tooth of blade by an incision (Figure 3o). Pretarsus well-developed, folded over the inner surface of the protibia, two-segmented; apical tarsomere elongated; pretarsal claws of unequal size (Figure 3p). Mesotibia with four, very small and blunt apical spines, strongly reflexed (Figure 2d; Figure 4m), surface smooth with sparse setae (Figure 4n). Mesotarsus smooth with sparse setae (Figure 4o), apex two-segmented; apical tarsomere elongated; mesotarsal claws of strongly unequal size (Figure 4p). Metatibia with four, very small and blunt apical spines, strongly reflexed (Figure 2h; Figure 5m), surface of metatibia smooth with sparse setae (Figure 5n). Metatarsus smooth with sparse setae (Figure 5o), apex two-segmented; apical tarsomere elongated; metatarsal claws of strongly unequal size (Figure 5p).

Measurements (Table 1): profemur length, 3.4–4.4 mm; protibia length, 3.3–4.5 mm; femoral tooth angle, 74.6–89.1°; number of femoral comb teeth, 7; distance between itf and ltf, 1.0–1.4 mm; pretarsus length, 2.1–3.2 mm; mesotibia length, 4.8–5.8 mm; mesotarsus length, 1.9–2.8 mm; number of mesotibial spines, 4; metatibia length, 4.6–6.0 mm; metatarsus length, 1.6–2.5 mm; number of metatibial spines, 4.

### 3.2. Principal Component Analysis

We explored relationships among species based on quantitative leg morphological data using PCA (Figure 6). The first two principal components (PCs) explained 84.89% of the total variance (Table 2). PC1 explained 61.74% of the variance in leg size (FL, FTL, FSL, MTL, MSL, HTL, and HSL). PC2 accounted for 23.14% of the data variability, of which indicators of the size of the femoral tooth (FA and DIF) were the significant variables for the grouping of species (Table 2). As shown in a PCA biplot, the OTUs for *C. atrata* and *H. maculaticollis* were grouped on the negative side of the PC 1 axis, while OTUs for *M. opalifera* and *P. kaempferi* were grouped on the positive side of the PC1 axis (Figure 6). Most OTUs for *M. opalifera* were positioned on the positive side of the PC2 axis using NFC. However, OTUs for *P. kaempferi* were positioned on the central to negative side of the PC2 axis using FA (Figure 6).

### 3.3. Taxonomic Key Based on Leg Morphology

1. Average femoral tooth angle above 80°; meso- and metatibia with four apical spines, spines strongly reflexed ------------------------------------------------- *Platypleura kaempferi* (Fabricius, 1794).

1. Average femoral tooth angle below 80°; meso- and metatibia with five to seven apical spines, spines not reflexed.

2. Profemur length 3.6–4.8 mm, femoral formula 2–1–8; surfaces of meso-, metatibia and meso-, metatarsus echinate ----------------------------------- *Meimuna opalifera* (Walker, 1850).

2. Profemur length 5.3–6.8 mm, femoral formula 2–1–7; surfaces of meso-, metatibia and meso-, metatarsus smooth.

3. Profemur surface echinate, distance between itf and ltf 1.4–1.8 mm, point of blade of tibia large and long, tooth-like; number of metatibial spines six or seven ---------------------------------------------------------------------- *Cryptotympana atrata* (Fabricius, 1775).

3. Profemur surface smooth, distance between itf and ltf 0.5–0.8 mm, point of blade of tibia very small or almost absent; number of metatibial spines five --------------------------------------------------------------------------- *Hyalessa maculaticollis* (Motschulsky, 1866).

### 3.4. Sequence Analysis and Phylogenetic Relationships

A 690 bp *CO1* region was successfully amplified and sequenced from 48 cicada samples (see Appendix for accession numbers; Table 3). Intraspecific genetic distances were 0.11% for *C. atrata*, 0.16% for *H. maculaticollis*, 0.03% for *M. opalifera,* and 0.21% for *P. kaempferi* (Table 3). Inter-specific genetic distances ranged from 17.70% to 20.16%, and was highest for *P. kaempferi* (20.16%) among the three species (Table 3). Additionally, we inferred the phylogenetic relationships among the four Korean cicada species using *CO1* sequences (Figure 7). The phylogenetic tree constructed using the neighbor-joining (NJ) method revealed that all samples from each species formed a monophyletic cluster with high bootstrap support (100%; Figure 7). Samples of *H. maculaticollis* and *M. opalifera* were more closely related to each other than to other species. Overall, these data suggest that the four cicada species are identifiable on the basis of sequence variation in the *CO1* region.

## 4. Discussion

We used multiple methodological approaches in combination to obtain reliable identification information for four cicada species, including morphological and ultrastructural features of the fore-leg (femur, femoral comb, protibial, and pretarsus), mid-leg (mesotibia and mesotarsus), and hind leg (metatibia and metatarsus), as well as *CO1* sequences of exuviae. Moreover, we developed an accurate taxonomic key and discussed variation in morphological characteristics in relation to molecular sequence data.

Most morphological studies of cicada exuviae have focused on size-related characteristics, mainly body length, head width, and wing length [23,24,27,28]. We found that qualitative characteristics, including ultrastructural features, as well as quantitative leg traits were valuable diagnostic characteristics for cicadas. Previously, Maccagnan and Martinelli [25] described the three leg pairs in detail and performed a comparative analysis of fifth-instar of cicadas in Brazil. Moreover, Midgley et al. [26] suggested that all three leg pairs should be included for detailed comparisons of cicadas. Thus, descriptions of all leg parts, including qualitative and quantitative characteristics, are essential for the accurate identification of the cicadas.

The fore-legs are expected to be closely related to the burrowing depth in the soil of nymphs and may provide promising characteristics for identification [2,3,28,29,30]. We were able to infer that fore-legs characteristics, such as the protibia and profemur surface patterns, might indeed be closely related to burrowing depth. *P. kaempferi*, which live 10 to 30 cm underground during the nymph stage [28,44], have a blunt protibia and smooth profemur, whereas *M. opalifera* have a sharp protibia and echinate profemur. Hou et al. [45] reported that *M. mongolica*, which is closely related to *M. opalifera*, could extend to 60 cm underground. *M. mongolica* also have very sharp and long protibia [28]. Thus, a sharp protibia and echinate profemur may be effective for deep digging. To verify this, observations of burrowing depth in *M. opalifera* nymphs and profemur surface pattern of *M. mongolica* are required. Moreover, ultrastructural patterns of the profemur were a stable characteristic, with consistent patterns at the species level. To evaluate the utility of profemur surface patterns for inferring phylogenetic relationships, further studies with an expanded and targeted sampling strategy based on recent phylogenetic analyses are necessary [46]. A morphological analysis revealed that FA and DIF were useful fore-leg characteristics for distinguishing among this studied species. FA and DIF could be effective diagnostic characteristics for the identification of cicadas based on the exuviae.

The number of the metatibial spines is a phylogenetically important characteristic accordingly to Hou et al. [28], who also reported that *C. atrata* has five spines on its hind legs. In contrast to the results of Hou et al. [28], we found that *C. atrata* has six to seven metatibial spines, and the variation in spine numbers can be explained by geographical or individual differences. According to this variation within species, phylogenetic classification based on the number of the metatibial spines needs to be reconsidered. The echinate surface pattern on the mid- and hind legs, only seen in *M. opalifera*, is useful for distinguishing among species. Thus, this unique pattern might be of taxonomic value, but further ultrastructural studies of other cicada species and comparative analyses of the exuviae and adult traits are needed to better understand their occurrence and taxonomic implications in cicadas. Furthermore, only *P. kaempferi* had four very small and strongly reflexed apical spines on the tibiae. The size, degree of reflex, and number of spines might be important diagnostic characteristics.

*COI* sequences have been proposed as an effective barcoding tool for animal identification [47,48,49] and have been successfully used for the identification of cicadas [46,50]. We found that *COI* sequences enabled accurate classification at the species level. Moreover, sequences obtained from the exuviae were identical to those obtained from adults in previous investigations [46,50]. These results clearly supported the efficiency and usefulness of genetic materials obtained from cicada exuviae, which are easy to collect and could be preserved in a range of environmental conditions [8].

Based on morphological and ultrastructural analyses of exuvial legs, we generated a taxonomic key for four major species in Korea. The observed morphological patterns, which were conserved at the species level, corresponded with phylogenetic relationships. These useful morphological and ultrastructural characteristics, such as profemur length and surface, femoral tooth angle, number of apical spines, and surface of meso- and metatibia of exuviae, may be informative for taxonomic and phylogenetic analyses. The extensive divergence of morphological and ultrastructural characteristics of the legs provides a basis for a taxonomic key for distinguishing among species. Moreover, the leg morphological and exuvial molecular characteristics, which are highly consistent at the species level, might be highly reliable identification markers.

## 5. Conclusions

In conclusion, we used statistical analyses and scanning electron microscopy to explore useful characteristics and ultrastructural features of cicada legs. Although broader sampling of species is necessary, these results provide a valuable framework for future studies of evolutionary trends and variation in cicada leg morphology. Moreover, our study demonstrates the importance of integrative studies for obtaining reliable information for species descriptions and for understanding the diversity of cicadas.

## Figures and Tables

**Figure 1 insects-10-00199-f001:**
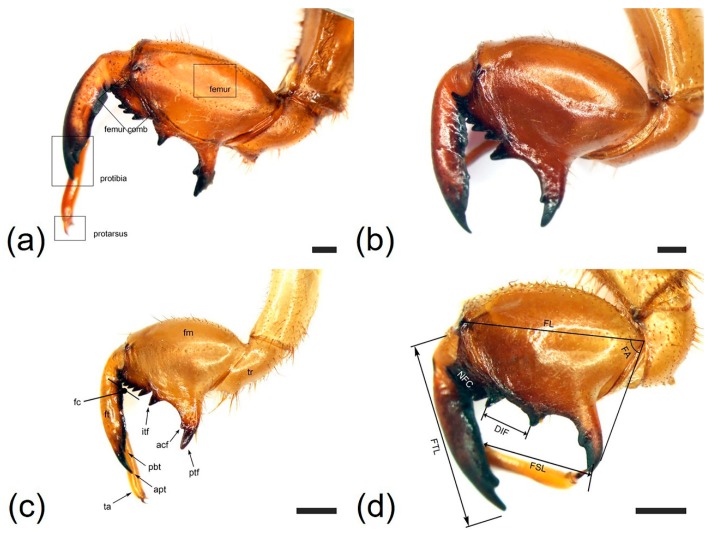
Morphological characteristics of the fore-legs of the four species, showing the measurements: (**a**) *Cryptotympana atrata*; (**b**) *Hyalessa maculaticollis*; (**c**) *Meimuna opalifera*; (**d**) *Platypleura kaempferi*. Open squares correspond to those of ultrastructural characteristics. acf, accessory tooth of femur; apt, apical tooth of tibia; fc, femoral comb; fm, femur; ft, tibia; itf, intermediate tooth of the femur; pbt, point of blade of tibia; ptf, posterior tooth of the femur; ta, tarsus; tr, trochanter; DIF, distance between the intermediate tooth of the femur and last tooth of the femoral comb; FA, femoral tooth angle; FL, profemur length; FSL, pretarsus length; FTL, protibia length; number of femoral comb teeth. All scale bars = 1.0 mm.

**Figure 2 insects-10-00199-f002:**
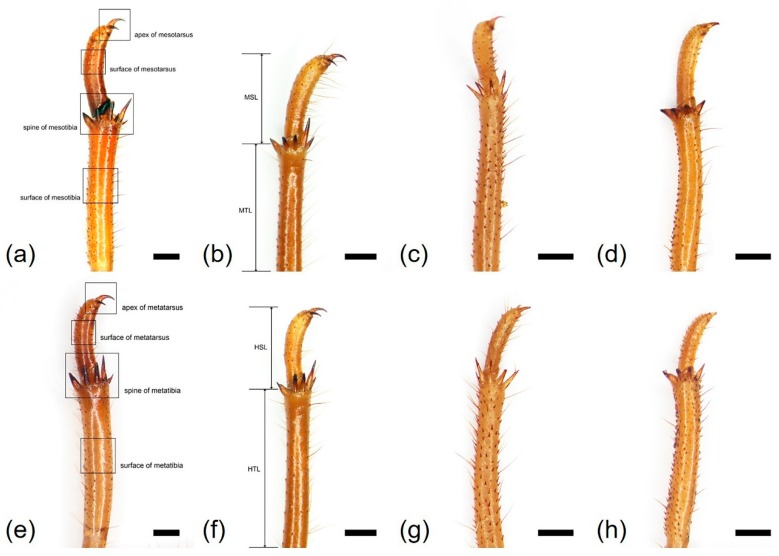
Morphological characteristics of the mid- and hind legs of the four species, showing the measurements: (**a**,**e**) *Cryptotympana atrata*; (**b**,**f**) *Hyalessa maculaticollis*; (**c**,**g**) *Meimuna opalifera*; (**d**,**h**) *Platypleura kaempferi*; (**a**–**d**) mid-legs; (**e**–**h**) hind legs. Open squares in (**a**) and (**e**) correspond to those of ultrastructural characteristics in Figure 4 and Figure 5, respectively. MSL, mesotarsus length; MTL, mesotibia length; HSL, metatarsus length; HTL, metatibia length. All scale bars = 1.0 mm.

**Figure 3 insects-10-00199-f003:**
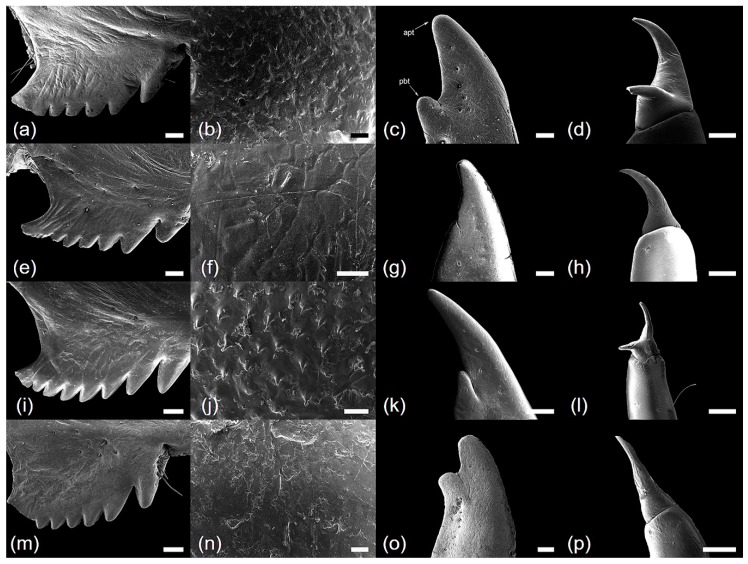
Scanning electron microscope micrographs of four fore-leg parts: (**a**–**d**) *Cryptotympana atrata*; (**e**–**h**) *Hyalessa maculaticollis*; (**i**–**l**) *Meimuna opalifera*; (**m**–**p**) *Platypleura kaempferi*; (**a**,**e**,**i**,**m**) femur comb; (**b**,**f**,**j**,**n**) center of femur; (**c**,**g**,**k**,**o**) protibia; (**d**,**h**,**l**,**p**) pretarsus. All scale bars = 200 μm (except **b**,**f**,**j**,**n**, scale bars = 20 μm).

**Figure 4 insects-10-00199-f004:**
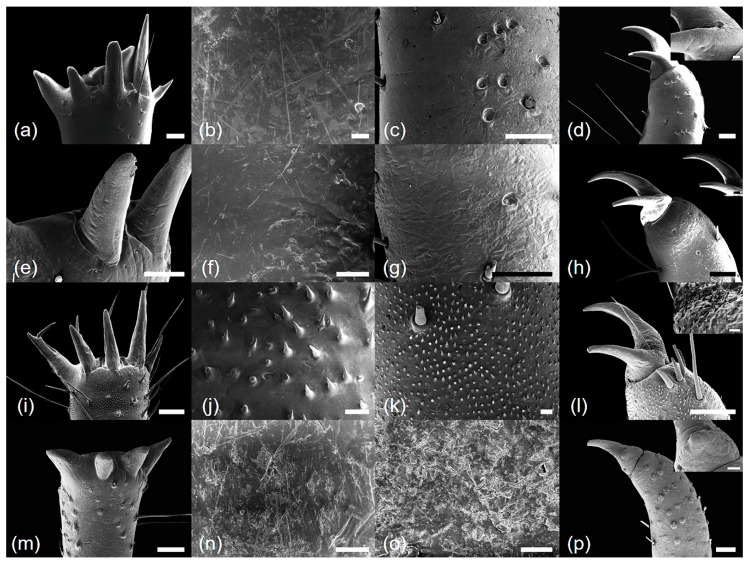
Scanning electron microscope micrographs of four mid-leg parts: (**a**–**d**) *Cryptotympana atrata*; (**e**–**h**) *Hyalessa maculaticollis*; (**i**–**l**) *Meimuna opalifera*; (**m**–**p**) *Platypleura kaempferi*; (**a**,**e**,**i**,**m**) spine of mesotibia; (**b**,**f**,**j**,**n**) surface of mesotibia; (**c**,**g**,**k**,**o**) surface of mesotarsus; (**d**,**h**,**l**,**p**) apex of mesotarsus. Inset micrographs show a surface pattern between their mesotarsal claws. All scale bars = 200 μm (except **b**, **f**, **j**, **k**, **n**, **o**, scale bars = 20 μm; inset **d**, **h**, **l**, and **p**, scale bars = 40, 200, 10, and 40 μm, respectively).

**Figure 5 insects-10-00199-f005:**
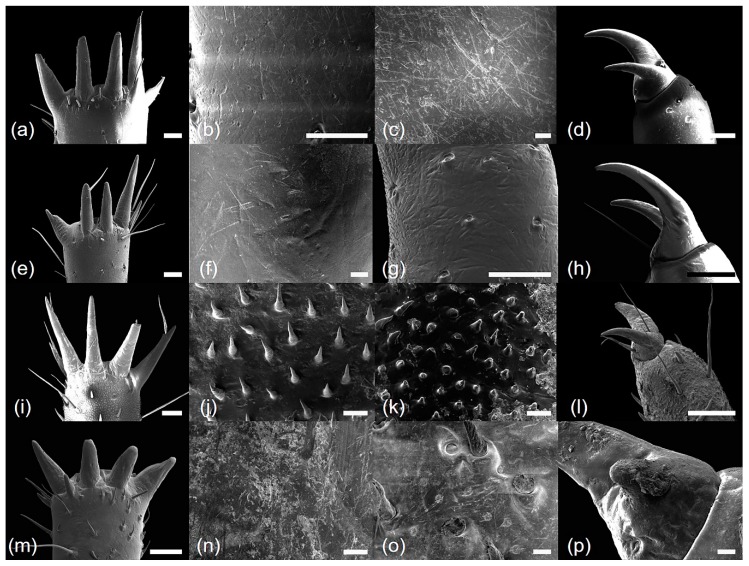
Scanning electron microscope micrographs of four hind leg parts: (**a**–**d**) *Cryptotympana atrata*; (**e**–**h**) *Hyalessa maculaticollis*; (**i**–**l**) *Meimuna opalifera*; (**m**–**p**) *Platypleura kaempferi*; (**a**,**e**,**i**,**m**) spine of metatibia; (**b**,**f**,**j**,**n**) surface of metatibia; (**c**,**g**,**k**,**o**) surface of metatarsus; (**d**,**h**,**l**,**p**) apex of metatarsus. All scale bars = 200 μm (except **b**, **c**, **f**, **j**, **k**, **n**, **o**, **p**, scale bars = 20 μm).

**Figure 6 insects-10-00199-f006:**
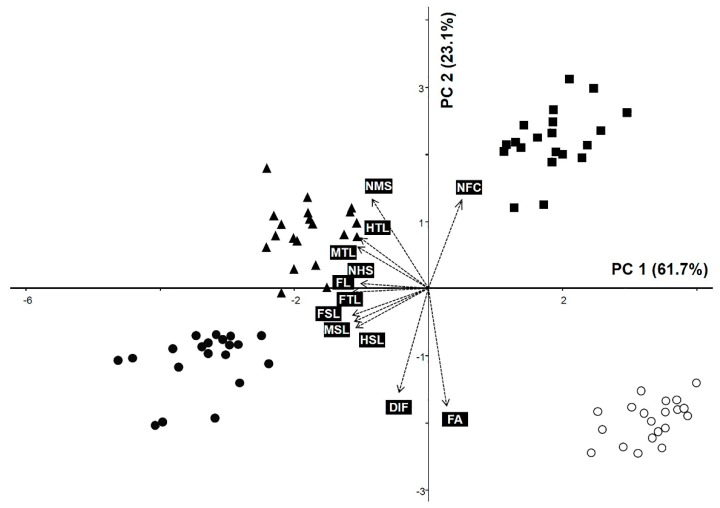
Principal component analysis (PCA) with twelve quantitative variables of cicada leg morphology for four species. FL, fore femur length; FTL, fore tibiae length; FA, femoral tooth angle; NFC, number of femoral comb teeth; DIF, distance between the intermediate tooth of the femur and last tooth of the femoral comb; FSL, pretarsus length; MTL, mesotibia length; MSL, mesotarsus length; NMS, number of mesotibial spines; HTL, metatibia length; HSL, metatarsus length; NHS, number of metatibial spines. filled circle, *Cryptotympana atrata*; filled triangle, *Hyalessa maculaticollis*; filled square, *Meimuna opalifera*; open circle, *Platypleura kaempferi*.

**Figure 7 insects-10-00199-f007:**
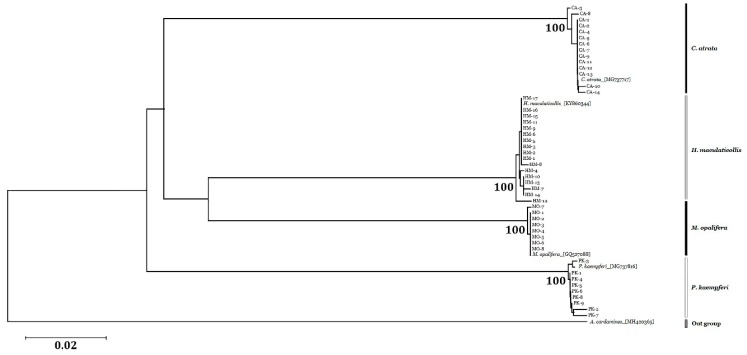
Phylogenetic tree using 48 samples of four species generated by the neighbor-joining method with 1000 bootstrap replicates based on *CO1* sequences. Scale bar represents 0.2 substitutions per site.

**Table 1 insects-10-00199-t001:** Surface patterns and measurements (mean ± S.D.) of fore-, mid-, and hind legs characteristics of four Cicadas species in Korea.

Characteristics	*Cryptotympana atrata*	*Hyalessa maculaticollis*	*Meimuna opalifera*	*Platypleura kaempferi*
*Fore-leg*				
Profemur surface	echinate	smooth	echinate	smooth
Profemur length (FL, mm)	6.18 ± 0.28	5.83 ± 0.31	4.28 ± 0.32	3.85 ± 0.26
Protibia length (FTL, mm)	6.09 ± 0.34	6.14 ± 0.26	4.04 ± 0.19	3.93 ± 0.29
Femoral comb surface	smooth	smooth	smooth	smooth
Femoral tooth angle (FA, °)	70.8 ± 4.26	67.1 ± 4.89	59.4 ± 3.85	82.8 ± 4.32
Number of femoral comb teeth (NFC)	(6)7(8)	(6)7	(7)8	7
Femoral formula	2–1–7	2–1–7	2–1–8	2–1–7
Distance between itf and ltf (DIF, mm)	1.56 ± 0.11	0.67 ± 0.08	0.55 ± 0.06	1.15 ± 1.00
Pretarsus length (FSL, mm)	4.23 ± 0.20	3.73 ± 0.22	2.42 ± 0.26	2.50 ± 0.29
*Mid-leg*				
Mesotibia surface	smooth	smooth	echinate	smooth
Mesotibia length (MTL, mm)	7.48 ± 0.26	7.92 ± 0.40	6.46 ± 0.61	5.35 ± 0.30
Number of mesotibia spine (NMS)	5	5	5	4
Mesotarsus surface	smooth	smooth	echinate	smooth
Mesotarsus length (MSL, mm)	3.94 ± 0.38	3.22 ± 0.33	2.21 ± 0.29	2.29 ± 0.23
*Hind leg*				
Metatibia surface	smooth	smooth	echinate	smooth
Metatibia length (HTL, mm)	7.61 ± 0.28	8.19 ± 0.54	6.62 ± 0.48	5.20 ± 0.37
Number of metatibia spine (NHS)	6–7	5	5	4
Metatarsus surface	smooth	smooth	echinate	smooth
Metatarsus length (HSL, mm)	3.37 ± 0.29	2.58 ± 0.19	1.98 ± 0.14	2.01 ± 0.21

**Table 2 insects-10-00199-t002:** Summary of the results of a principal component analysis of the twelve quantitative leg morphological characteristics of four species of Korean cicadas.

No.	Characteristics	PC 1	PC 2	PC 3
		(61.743%)	(23.144%)	(7.512%)
1	FL	**−0.3579**	0.0004	−0.1003
2	FTL	**−0.3450**	−0.0208	−0.2532
3	FA	0.0829	**−0.536**	−0.1981
4	NFC	0.1532	0.4048	**0.4666**
5	DIF	−0.1326	**−0.4735**	0.4615
6	FSL	**−0.3464**	−0.1239	−0.0448
7	MTL	**−0.3216**	0.1876	−0.2463
8	MSL	**−0.3378**	−0.1538	0.0779
9	NMS	−0.2571	**0.4051**	0.1295
10	HTL	**−0.3107**	0.2297	−0.2421
11	HSL	**−0.3309**	−0.181	0.2019
12	NHS	−0.3087	0.0203	**0.5205**
	Eigenvalue	7.409	2.777	0.901

**Table 3 insects-10-00199-t003:** Statistical characteristics of *CO1* regions.

Species	Amplicon Length (bp)	^1^ Intraspecific Distance (%)	^1^ Interspecific Distance (%)	G + C (%)
*C. atrata*	690	0.11 ± 0.13	19.67 ± 0.80	31.00
*H. maculaticollis*	690	0.16 ± 0.17	18.60 ± 1.65	30.87
*P. kaempferi*	690	0.21 ± 0.20	20.16 ± 0.07	30.13
*M. opalifera*	690	0.03 ± 0.07	17.72 ± 1.59	30.84

^1^ (mean ± S.D.).

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
