# Peer review of "Comparative Morphological, Ultrastructural, and Molecular Studies of Four Cicadinae Species Using Exuvial Legs"

_insects, 2019, doi:10.3390/insects10070199_

Reviewer 1 Report

Comments to authors (Song et al., Insects)

Authors performed morphological analyses on exuviae of four species of Korean Cicada in order to identify species only from exuviae. The reasons why they focused on exuviae were interestingly introduced. As results, four species can be clearly distinguished by both of morphology and molecular sequencing analyses. All of analyses were logical and well performed. Results were clearly displayed.

The manuscript length is also appropriate, compactly written. I highly recommend this manuscript to be published in the journal.

Minor comments,

Please add adult photo and overview of exuviae of four species, maybe as Figure 1, which help readers to know their material species easily.

Are there any other sympatric cicada species potentially confused with focal four species? Please describe such information (around Line 62-66 might be good).

Line 65-66 “Moreover, Cicadidae Periostracum can potentially be adulterated with the exuviae of species other than C. atrata.” I could not understand what authors want to state in this sentence.

Author Response

Response to Reviewer 1 Comments

English language and style

 ( )  Extensive editing of English language and style required

 ( )  Moderate English changes required

 (x)  English language and style are fine/minor spell check required

 ( )  I don't feel qualified to judge about the English language and style

Authors performed morphological analyses on exuviae of four species of Korean Cicada in order to identify species only from exuviae. The reasons why they focused on exuviae were interestingly introduced. As results, four species can be clearly distinguished by both of morphology and molecular sequencing analyses. All of analyses were logical and well performed. Results were clearly displayed.

The manuscript length is also appropriate, compactly written. I highly recommend this manuscript to be published in the journal.

- Thank you very much for your kind words about our paper.

Point 1: Please add adult photo and overview of exuviae of four species, maybe as Figure 1, which help readers to know their material species easily.

Response 1: Thank you for this suggestion. We fully agree with your suggestions. However, the object of this study is to, focusing on the exuviae, provide detailed morphological and ultrastructural exuvial legs and their DNA sequence data. Moreover, we have already lots of figures. Thus, we did not include adult photos.  

Point 2: Are there any other sympatric cicada species potentially confused with focal four species? Please describe such information (around Line 62-66 might be good).

Response 2: Although Meimuna mongolica, which is closely related to M. opalifera, is sympatric species, they are easily distinguished based on the exuvial size. Moreover, thirteen Korean cicada species are almost different genera. Thus, we did not have to provide potentially confused species information.

Point 3: Line 65-66 “Moreover, Cicadidae Periostracum can potentially be adulterated with the exuviae of species other than C. atrata.” I could not understand what authors want to state in this sentence.

Response 3: We really appreciate this comment. Following your comments, we have rewritten this sentence more clearly.

Revised: Cicadidae Periostracum, which is exuviae of C. atrata, as an insect medicinal source can potentially be adulterated and misused with the exuviae of these dominantly distributed species.

Reviewer 2 Report

The paper of Song et al reports data of a study carried out on Cicadas exuviae, combining different methods to achieve the goal of discriminating different species. The paper is written in a correct form. Pictures are clear and presented in an appropriate way. I have appended below some comments to be taken into account by the authors.

Line 76: did you actually wait for nymphs to become adults, and then you related the resulting exuvia to the newly emerged adult. If so, please explain better.

Line 87: it seems that you did not discriminate between nymphs leading to males and nymphs leading to females in all the investigated species. Since in insects it is quite common that males and females show differences in size, I think that you should have considered this, especially in the morphometric analysis.

Line 90: pretarsus will be more appropriate than protarsus, please change throughout the text.

Line 93: This is far too generic, which tool did you actually used? Please specify whether you used a specific software or image analysis package.

Line 379-384: it is not clear to me why COI sequence from exuviae should differ from COI sequences from adults of the same species. It also looks to me that morphological analysis is more than enough to characterise the different exuvial nymphs, why one should do molecular analysis when the morphology already gives you the correct answer?

In figure legend of figure 3, 4 and 5 there are no correct pointing of the letters to the actual pictures (see for example in figure 3 E-G should be E-H), please check and correct.

As regards SEM pictures, please use scale bar slightly larger (although not uniform in size), some of them are too small. Pleas also avoid labelling with descriptions like in figure 3 a, b, c, and d , figure 4 a, b, c and d and figure 5 a, b, c and d. Information is already on the legend.

In figure 4 d, h, l and p you added inset pictures, these need to be added in the legend of figures and you need to add bar scale to them.

Author Response

Response to Reviewer 2 Comments

Point 1: English language and style

 ( )  Extensive editing of English language and style required

 ( )  Moderate English changes required

 (x)  English language and style are fine/minor spell check required

 ( )  I don't feel qualified to judge about the English language and style

The paper of Song et al reports data of a study carried out on Cicadas exuviae, combining different methods to achieve the goal of discriminating different species. The paper is written in a correct form. Pictures are clear and presented in an appropriate way. I have appended below some comments to be taken into account by the authors.

Point 1: Line 76: did you actually wait for nymphs to become adults, and then you related the resulting exuvia to the newly emerged adult. If so, please explain better.

Response 1: As you already know, Cicadas emerge from the ground for the final molt and climb up plant stems and trunks. They hang on the underside of a leaf or on a branch or trunk and begin to molt. Thus, exuviae from the final instars are found on tree branches or leaves where the final molt takes place. We collected from the branch or trunk of trees, i.e., Retusa fringetree, oriental flowering cherry, Sawleaf zelkova and other kinds of trees. To ensure positive species identification, we also compare the CO1 sequence data using barcoding.

Revised: Exuviae were collected from branch or trunk of trees, i.e., Chionanthus retusus Lindl. & Paxton (Retusa fringetree), Prunus serrulata f. spontanea (E. H. Wilson) C.S. Chang (Oriental flowering cherry), Zelkova serrata (Thunb.) Makino (Sawleaf zelkova) and other kinds of trees in Korean natural populations after moulting from July to August, 2018.

Point 2: Line 87: it seems that you did not discriminate between nymphs leading to males and nymphs leading to females in all the investigated species. Since in insects it is quite common that males and females show differences in size, I think that you should have considered this, especially in the morphometric analysis.

Response 2: Yes, we fully agree with your suggestions. However, according to our previous investigation, which is ‘Studies on morphological characteristics of the original species, Cryptotympana atrata as a Cicadidae Periostracum (Song et al., Korean Herb. Med. Inf., 2019)’, significant exuvium size differences were not found between the sexual morphs including size of leg morphology. In addition, there were no differences in morphological and micromorphological characteristics between females and males except the presence of gonapophysis in female only. Thus, we focused on the quantitative variations and differences between interspecies, not sexual dimorphism.

Point 3: Line 90: pretarsus will be more appropriate than protarsus, please change throughout the text.

Response 3: We really appreciate this comment. We fully changed terminology from protarsus to pretarsus in the all text as your recommendation.

Point 4: Line 93: This is far too generic, which tool did you actually used? Please specify whether you used a specific software or image analysis package.

Response 4: We provided the specific image analysis package.

Revised: Most of these measurements were obtained using simple measurement tools (Olympus DP21 digital photography system, Olympus, Tokyo, Japan) based on images obtained using a digital camera (Olympus DP21) for microscopes.

Point 5: Line 379-384: it is not clear to me why COI sequence from exuviae should differ from COI sequences from adults of the same species. It also looks to me that morphological analysis is more than enough to characterise the different exuvial nymphs, why one should do molecular analysis when the morphology already gives you the correct answer?

Response 5: Yes, we agree with your opinions. However, we think that both morphological and COI sequence analysis are needed.

First, Exuviae of Cryptotympana atrata are commonly used in traditional medicine in Asia, including Korea, China, and Taiwan. However, Cicadidae Periostracum, which is exuviae of C. atrata was currently distributed with mixtures (e.g., counterfeits, adulterants), crumbly, or powder forms in commercial markets. The counterfeits, adulterants and their medicinal materials of poor quality degrade the clinical effects of medicine and may even result in death of patients. Thus, authentication is a critical step for successful and reliable clinical applications. Besides, for the quality control of these insect medicinal materials, authentication and standardization are absolutely needed. Especially, we can’t discriminate and identify this material when dried exuviae are marketed to crumbly or powder forms. For this reason, we tried to establish the various methods of identification and authentication for accurate species identification with just a fraction of the medicinal materials. In this study, we suggested the authentication and identification technique of exuviae of C. atrata using various methodologies such as morphological, micromorphological, and barcoding analyses.

Second, Exuviae have been demonstrated to be reliable genetic sources for a variety of species, including honey bees, mosquitoes, scarabs, dragonflies, and tarantulas, etc. Thus, we tried to confirm the possibility of cicada exoskeleton as a reliable genetic source.

Third, COI sequence from exuviae should same from COI sequences from adults of the same species, of course. Thus, we wanted to ensure an accurate the identification of Cicada species using only exuviae by using morphological and molecular data.

Point 6: In figure legend of figure 3, 4 and 5 there are no correct pointing of the letters to the actual pictures (see for example in figure 3 E-G should be E-H), please check and correct.

Response 6: At first, we made a mistake. Following your comments, we corrected figure legend. Moreover, we checked all figure legends and changed lowercase characters from uppercase characters in the text.

Revised: Figure 3. Scanning electron microscope micrographs of four fore-leg parts. (a–d) Cryptotympana atrata; (e–h) Hyalessa maculaticollis; (i–l) Meimuna opalifera; (m–p) Platypleura kaempferi; (a, e, i, m) Femur comb; (b, f, j, n) Center of femur; (c, g, k, o) Protibia; (d, h, l, p) Protarsus. All scale bars = 200 μm (except b, f, j, n, scale bars = 20 μm).

Point 7: As regards SEM pictures, please use scale bar slightly larger (although not uniform in size), some of them are too small. Pleas also avoid labelling with descriptions like in figure 3 a, b, c, and d, figure 4 a, b, c and d and figure 5 a, b, c and d. Information is already on the legend.

Response 7: We fully agree with your suggestions. Following your comments, we used more larger scale bars in all figures. Moreover, we deleted the labelling with descriptions like in figure 3 a, b, c, and d, figure 4 a, b, c, and d and figure 5 a, b, c and d.

Point 8: In figure 4 d, h, l and p you added inset pictures, these need to be added in the legend of figures and you need to add bar scale to them.

Response 8: Thank you for this suggestion. Following your comments, we added the information of inset micrographs and their scales.

Revised: Inset micrographs show a surface pattern between their mesotarsal claws. All scale bars = 200 μm (except b, f, j, k, n, o, scale bars = 20 μm; inset d, h, l, and p, scale bars = 40, 200, 10, and 40 μm, respectively).
